# Antibody–Drug Conjugate to Treat Meningiomas

**DOI:** 10.3390/ph14050427

**Published:** 2021-05-02

**Authors:** Kai Chen, Yingnan Si, Jianfa Ou, Jia-Shiung Guan, Seulhee Kim, Patrick Ernst, Ya Zhang, Lufang Zhou, Xiaosi Han, Xiaoguang (Margaret) Liu

**Affiliations:** 1Department of Biomedical Engineering, University of Alabama at Birmingham (UAB), 1825 University Blvd, Birmingham, AL 35294, USA; kaisdzb@uab.edu (K.C.); yingnan@uab.edu (Y.S.); jou@uab.edu (J.O.); seulheekim@uabmc.edu (S.K.); yazhang9@uab.edu (Y.Z.); lfzhou@uab.edu (L.Z.); 2Department of Medicine, University of Alabama at Birmingham, 703 19th Street South, Birmingham, AL 35294, USA; guan0926@uab.edu (J.-S.G.); pernst@uab.edu (P.E.); 3Department of Neurology, University of Alabama at Birmingham, 1824 6th Avenue South, Birmingham, AL 35294, USA; xhan@uabmc.edu; 4O’Neal Comprehensive Cancer Center, University of Alabama at Birmingham, 1824 6th Avenue South, Birmingham, AL 35233, USA

**Keywords:** meningiomas, targeted therapy, somatostatin receptor 2, monoclonal antibody, antibody–drug conjugate

## Abstract

Meningiomas are primary tumors of the central nervous system with high recurrence. It has been reported that somatostatin receptor 2 (SSTR2) is highly expressed in most meningiomas, but there is no effective targeted therapy approved to control meningiomas. This study aimed to develop and evaluate an anti-SSTR2 antibody–drug conjugate (ADC) to target and treat meningiomas. The meningioma targeting, circulation stability, toxicity, and anti-tumor efficacy of SSTR2 ADC were evaluated using cell lines and/or an intracranial xenograft mouse model. The flow cytometry analysis showed that the anti-SSTR2 mAb had a high binding rate of >98% to meningioma CH157-MN cells but a low binding rate of <5% to the normal arachnoidal AC07 cells. The In Vivo Imaging System (IVIS) imaging demonstrated that the Cy5.5-labeled ADC targeted and accumulated in meningioma xenograft but not in normal organs. The pharmacokinetics study and histological analysis confirmed the stability and minimal toxicity. In vitro anti-cancer cytotoxicity indicated a high potency of ADC with an IC_50_ value of <10 nM. In vivo anti-tumor efficacy showed that the anti-SSTR2 ADC with doses of 8 and 16 mg/kg body weight effectively inhibited tumor growth. This study demonstrated that the anti-SSTR2 ADC can target meningioma and reduce the tumor growth.

## 1. Introduction

Meningiomas are the most common primary tumors in the central nervous system. Malignant meningiomas have overall five-year survival rates of 64.6–68.5% due to the greater recurrence rate and mortality as reported by the Central Brain Tumor Registry of the United States (CBTRUS) 2020 [1]. A significant subset of meningiomas (WHO grade II and III) have aggressive features and are associated with uncontrollable growth and high rates of morbidity and mortality [2,3]. The tumors contain widespread genomic disruption (amplification, deletion, and rearrangement), inactivation of neurofibromatosis type 2 (NF2) tumor suppressor gene, and other mutations [4,5]. 

Somatostatin receptors (SSTRs), especially the SSTR2a subtype, are highly expressed in 90% of meningiomas [6,7,8], although the functional role of high SSTR density remains unclear [6]. Clinical studies demonstrated that SSTR2 is an ideal surface receptor to target meningiomas [7,9,10]. Clinicians have been using radio-labeled octreotide (a somatostatins analog), e.g., gallium-68 (^68^Ga)-DOTATATE [7,11], as a brain scintigraphy tracer to delineate the extent of meningiomas and to pathologically define extra-axial lesions. In addition to diagnosis and detection [12,13], somatostatin analog, in combination with other systemic therapy, such as everolimus/octreotide, has demonstrated the ability to inhibit meningioma tumor growth [14,15,16], although further investigation is needed [8,17,18]. Despite these achievements, it is imperative to develop a targeted therapy to treat SSTR2-positive low- and high-grade meningiomas.

Antibody–drug conjugates (ADCs) have been developed to treat cancers or tumors [19,20,21,22] and have demonstrated promising clinical efficacy and minimal adverse effects [23,24,25,26]. For instance, the gemtuzumab ozogamicin (anti-CD33 mAb-N-acetyl gamma calicheamicin conjugate) [27,28] for acute myeloid leukemia treatment, traztuzumab deruxtecan (anti-HER2 mAb-topoisomerase I inhibitor) [29] and trastuzumab emtansine (anti-HER2 mAb-Mertansine DM1 conjugate) [30,31] for chemotherapy refractory or advanced HER2-positive breast cancer, brentuximab vedotin (anti-CD30 mAb-MMAE conjugate) [32,33] for relapsed Hodgkin lymphoma, and ^131^I-Tositumomab (anti-CD20 mAb-Iodine 131) [34] and ^90^Y-Ibritumomab tiuxetan (anti-CD20 mAb-Yttrium-90) [35] for non-Hodgkin lymphoma have been approved by the FDA and used in clinics. Compared to standard chemotherapies, ADCs can specifically target cancer cells, deliver highly cytotoxic payloads, and reduce adverse effects. To our best knowledge, ADC therapy has not been developed for meningioma treatment so far. In addition to mAb and ADC, the nanobody (nAb) [36] and nAb–drug conjugates, such as FDA-approved abraxane [37], also showed great therapeutic potentials for cancer treatment. 

The objective of this study was to develop and evaluate an ADC-based targeted therapy to treat aggressive meningiomas overexpressing SSTR2. Our mAb that targets the extracellular domain of surface SSTR2 was produced in fed-batch culture, purified using liquid chromatography, and conjugated with an FDA-approved potent payload, mertansine (DM1), which inhibits the polymerization of microtubules and cell proliferation [30,31,38]. The meningioma-specific targeting, pharmacokinetics, toxicity, and anti-tumor efficacy of the constructed ADC were evaluated in vitro using cell lines or in vivo using an intracranial xenograft mouse model. Our study showed that the anti-SSTR2 ADC can effectively target meningioma and inhibit the tumor proliferation with minimal toxicity. 

## 2. Results and Discussion

### 2.1. Anti-SSTR2 ADC Construction

In this study, we constructed and evaluated the SSTR2-targeting ADC for meningioma treatment using our previously developed anti-human and mouse SSTR2 mAbs (IgG1 kappa) that targets the extracellular domain of SSTR2 [39]. We evaluated four mAb clones and the top clone that had high SSTR2 affinity (equilibrium dissociation constant of 6.6 nM) and meningioma specificity (binding rate >90%) was identified and produced in a stirred-tank bioreactor with a final titer of 100 mg/L in fed-batch bioproduction, as described in Materials and Methods. The produced mAb was conjugated with DM1 via sulfo-SMCC linker to construct ADC (Figure 1A) following our previously established ADC construction platform [38]. HPLC analysis showed that the average drug-to-antibody ratio (DAR) of the constructed ADC was 4.6. After i.v. administration, ADC circulates through the bloodstream and targets meningioma by binding the overexpressed surface receptor SSTR2. After surface binding, the ADC is internalized into the cytoplasm of meningioma cells through receptor-mediated endocytosis to form a late endosome, and free drug is released via lysosomal degradation [40]. Finally, the potent DM1 depolymerizes microtubulin, induces apoptosis and programmed cell death [41,42], and inhibits tumor cell proliferation (Figure 1B). 

### 2.2. High Surface Binding to Meningioma

Flow cytometry analysis was performed to assess the in vitro meningioma-targeting specificity of our anti-SSTR2 mAb. As presented in Figure 2A, the AF647-labeled mAb showed high surface binding to meningioma cell line CH157-MN (99.6%) and low binding to normal arachnoidal cell line AC07 (1.48%). Furthermore, we evaluated the in vivo meningioma specificity and biodistribution of anti-SSTR2 mAb and ADC using a CH157-MN-FLuc intracranially xenografted mouse model. As described in Figure 2A,B, live-animal IVIS imaging demonstrated that the bioluminescent signal (FLuc) and fluorescent signal (Cy5.5) well overlapped, indicating that mAb and ADC effectively targeted and accumulated in meningioma xenograft within 24 h post intravenous (i.v.) injection. These results also confirmed that the DM1 conjugation at the lysine residue did not change the antigen-binding capability of ADC. The non-specific targeting in normal organs, such as brain, heart, lung, kidney, and spleen, was not detected in ex vivo IVIS imaging (Figure 2D). As we reported before [39], anti-SSTR2 mAb has no cross reactivity among SSTR1–5 while it targets both human SSTR2 and mouse SSTR2. The in vivo targeting images in human meningioma xenografted mice indicated that anti-SSTR2 ADC can specifically target meningioma and effectively deliver the conjugated cargos (cyanine-5.5 or DM1).

### 2.3. In Vitro ADC Cytotoxicity Studies

The in vitro anti-meningioma cytotoxicity of ADC and free drug DM1 was tested with CH157-MN and AC07 cells in 96-well plates. Treatment with eight doses of DM1 in a three-day assay, including 0.1, 0.5, 2, 10, 30, 60, 100, and 300 nM, reduced cell viability to 24.82%, 7.09%, 6.25%, 3.32%, 2.03%, 0.83%, 0.80%, and 0.72% for CH157-MN and 26.88%, 11.12%, 7.65%, 3.99%, 2.87%, 2.05%, 1.84%, and 1.37% for AC07 (Figure 3A). The calculated IC_50_ value of DM1 was 0.31 nM for CH157-MN and 0.56 nM for AC07, which confirmed that DM1 is a highly potent cytotoxin without cancer or tumor cell selectivity [30,31]. Treatment with nine doses of ADC, including 0.5, 1, 2, 10, 25, 50, 100, 250, and 500 nM, decreased the final viability of CH157-MN to 95.70%, 96.48%, 96.30%, 41.26%, 18.90%, 11.80%, 10.35%, 7.96%, and 2.67% (Figure 3A) with an IC_50_ of 7.41 nM. These results reveal that SSTR2 ADC had a high cytotoxicity to meningioma cells. 

In addition to DM1 and ADC, we also tested the possible cytotoxicity of mAb, somatostatin (SST) analogue, and octreotide. The three-day in vitro assay showed that neither anti-SSTR2 mAb (2 µM) nor octreotide (2 µM) caused cytotoxicity in CH157-MN as compared to the PBS control (Figure 3B). All together, the specificity study and in vitro cytotoxicity study indicated that the anti-SSTR2 mAb can effectively target meningioma and deliver the highly potent payload DM1.

### 2.4. Pharmacokinetics (PK)

The ADC was i.v. injected into NSG mice at doses of 10, 15, 20, and 25 mg/kg in the PK study. Approximately 10–15 µL serum samples were collected from a tail nick at 0.5, 2, 7, 24, 48, and 72 h post injection. The kinetic profile of the serum titers of ADC is presented in Figure 4A. The PK modeling described in Materials and Methods was performed to analyze typical PK parameters to guide the in vivo anti-meningioma study. Specifically, the calculated area under the curve (AUC) was 58.82–140.95 µg day/mL, half-life t_1/2_ was 1.67–2.27 days, recommended dose D was 8.06–17.96 mg/kg, and recommended dosing interval τ was 3.99–4.91 days (Figure 4B). The other parameters were volume of distribution V_d_ of 76.80–79.28 mL/kg, clearance C_L_ of 24.16–31.70 mL/day/kg, and bioavailability F of 13.62–18.55%. Considering that anti-SSTR2 ADC targets and accumulates in meningioma tumor within 24 h post i.v. administration (Figure 2B), the half-life t_1/2_ of 1.67–2.27 days indicated a high circulation stability. Furthermore, the HPLC analysis did not detect cleaved DM1 and also confirmed the structural integrity and stability of injected ADC. Moreover, the calculated D and τ suggested the treatment strategies of doses of 8 and 16 mg/kg with an administration interval of 3 days in the following in vivo anti-meningioma animal study. The PK parameters of Lutathera have been reported as AUC of 41 ng h/mL, half-life t_1/2_ of 3.5 h, C_max_ of 10 ng/mL, and C_L_ of 4.5 L/h in adults [43] (https://reference.medscape.com/ (accessed on 26 April, 2021)), and AUC of 45.11–67.02 µg min/mL, t_1/2_ of 19.6–24.4 min, C_max_/Dose of 29.4–38.0 in rats (assessment report of the Europe Medicines Agency). As compared to Lutathera, the ADC has higher plasma stability. 

### 2.5. In Vivo Anti-Meningioma Efficacy

The NSG mice carrying CH157-MN-FLuc xenografts were i.v. administrated with 8 mg/kg anti-SSTR2 mAb (control), 8 mg/kg ADC, or 16 mg/kg ADC in three groups. We started ADC treatment when an obvious (i.e., >1000) bioluminescent signal was detected in IVIS imaging on Day 9 post cells implantation. The xenograft mice with similar bioluminescence signals (i.e., tumor volume) were randomized into three groups (*n* = 4) for treatment. The tumor volume was monitored by measuring fluorescent flux using IVIS imaging. Figure 5 shows that meningioma tumor volume was significantly reduced by 84–88% in ADC treatment groups compared to the mAb control group (*p* ≤ 0.005). The total bioluminescent radiance intensities in the meningioma tumor (ROI) were 25.4 ± 3.7 (21.0–28.5), 3.8 ± 2.0 (1.3–5.9), and 3.3 ± 2.9 (1.1–6.5) × 10^5^ photons/sec/cm^2^/sr for mAb (control), 8 mg/kg ADC, or 16 mg/kg ADC, respectively. Treatment was terminated when the control group showed obvious slow locomotion and body weight loss (>20%) on Day 15. These in vivo data indicate that anti-SSTR2 ADC can effectively control the tumor growth of aggressive meningioma.

### 2.6. Toxicity Evaluation

To evaluate the potential toxicity of ADC, we injected PBS, 16 mg/kg mAb, and 16 mg/kg ADC into NSG mice (*n* = 6). The body weight of mice was monitored daily for 21 days and, as expected, no obvious difference among the three groups was observed (Figure 6A). Moreover, there were no overt changes in general health, including water intake, breathing, and locomotion. At the end of the study, mice were sacrificed to collect major organs, such as brain, lung, heart, kidney, liver, and spleen, for further toxicity analysis via H&E staining. Pathologic assessment of H&E-stained organ sections did not show any signs of acute or chronic inflammation or apoptotic or necrotic regions in the PBS control group (not shown), mAb group (Figure 6B), and ADC group (Figure 6C). The Human Atlas Project reported high-level SSTR2 mRNA in brain, but the H&E-stained brain tissue did not show morphology changes or necrosis. Furthermore, the IHC staining of human cerebellum and cerebrum slides with our anti-SSTR2 mAb did not detect non-specific binding (Figure 6D), which was consistent with the flow cytometry analysis data (Figure 2A). Considering that ADC is a dose-dependent targeted therapy (Figure 3A), the toxicity caused by possible off-target ADC could be minimal. The anti-SSTR2 ADC had no obvious off-target effects on body weight, overall survival, and major organs, which indicates that it is a safe targeted therapy for SSTR2-positive meningioma. 

The standard systemic therapies, such as sunitinib and everolimus/octreotide, have been evaluated to control aggressively recurrent meningiomas, but their clinical efficiency is poor [14,15,16]. More recently, the Lutathera combining [^177^Lu]Lu-DOTA-TATE with [^68^Ga]Ga-DOTA-TATE or [^68^Ga]Ga-DOTA-TOC DOTA-(D-Phe1, Tyr3)-octreotide is being evaluated in clinical trials for meningioma treatment [11,44,45]. The SST analogue octreotide targets meningioma and ^177^Lu damages DNA. The clinical trial data showed a median overall survival of 17.2 months in grade III patients (*n* = 8) and did not reach a median follow-up of 20 months in grade I (*n* = 5) and II (*n* = 7) patients [11]. In addition to limited clinical efficiency, the short radiopharmaceutical shelf life and decay of ^177^Lu causing active concentration changes also hampers its clinical application. Compared to Lutathera, the anti-SSTR2 mAb-based ADC has the advantages of a long shelf life, plasma stability, targeting specificity, and high anti-tumor efficacy. Compared to standard chemotherapies [46], the anti-SSTR2 ADC shows minimal side effects and high cancer or tumor specificity. 

## 3. Materials and Methods

### 3.1. Cell Lines, Seed Cultures, and Media

Human meningioma cell lines, including malignant CH157-MN (kindly provided by Professor Yancey Gillespie at the University of Alabama at Birmingham, Birmingham, AL, USA) and CH157-MN-FLuc (generated in our lab by overexpressing FLuc in CH157-MN) were maintained in DMEM/F12 (Gibco, Grand Island, NY, USA) supplemented with 10% fetal bovine serum (FBS) in T25 or T75 flasks. The normal arachnoidal cell line AC07 (kindly provided by Professor Vijaya Ramesh at Harvard University, Cambridge, MA, USA) was maintained in DMEM with 15% FBS in T25 or T75 flasks as control cells. The anti-SSTR2 mAb-producing hybridoma cells were cultivated in Hybridoma-SFM with 4 mM L-glutamine in SF125 shaker flasks with agitation speed of 130 rpm. All these cultures were incubated at 37 °C and 5% CO_2_ in a humidified incubator (Caron, Marietta, OH, USA). The viable cell density (VCD) and viability were measured using a Countess II automated cell counter (Fisher Scientific, Waltham, MA, USA) or trypan blue. All basal media, supplements, and reagents used in this study were purchased from Fisher Scientific unless otherwise specified. 

### 3.2. Mice and Intracranial Xenograft Model

The five-week-old NSG (NOD.Cg-Prkdc<scid> Il2rg<tm1Wjl>/SzJ) or Nude (J:NU) male and female mice (equal number) were purchased from Jackson Laboratory (Bar Harbor, ME, USA). Approximately 0.2 × 10^5^ CH157-MN-FLuc cells in 3 µL of PBS were stereotactically implanted into the frontal region of the cerebral cortex, 2 mm lateral, 1 mm anterior, and 1.5 mm ventricle of bregma, at a rate of 0.4 µL per minute. The Stoelting Just for Mouse™ Stereotaxic Instrument equipped with a Cordless Micro Drill, Quintessential Stereotaxic Injector, and Hamilton™ 1700 Series Gastight™ Syringes (Thermo Fisher Scientific, Waltham, MA, USA) was used for intracranial xenograft mice model generation. 

### 3.3. Anti-SSTR2 mAb and ADC Generation

The SSTR2 mAb was produced using hybridoma cells in a 2-L stirred-tank bioreactor (Chemglass, Vineland, NJ, USA), controlled at 37 °C, pH 7.0, DO 50%, and agitation 70 rpm [39,47]. The bioreactor was seeded with hybridoma cells at a VCD of 0.3 × 10^6^ cells/mL in Hybridoma-SFM basal medium supplemented with 4 g/L glucose, 6 mM L-glutamine, and 3.5 g/L Cell Boost #6 on Day 0. Fed-batch production was performed by feeding 4 g/L glucose, 6 mM L-glutamine, and 3.5 g/L Cell Boost #6 on Day 3. The anti-SSTR2 mAb was purified using an NGC liquid chromatography system (Bio-Rad, Hercules, CA, USA) equipped with Protein A and ion exchange columns [38,48]. The ADC was constructed by conjugating DM1 with purified anti-SSTR2 mAb via sulfo-SMCC linker following our previously developed platform [38,49]. ADC product was concentrated and purified using 10 kDa MWCO concentrator (Fisher) to remove most linker and free drugs first. Then, a PD SpinTrap^TM^ G25 column (GE Healthcare, Chicago, IL, USA) was applied to remove the chemicals used in conjugation. Finally, high-performance liquid chromatography (HPLC, Shimadzu, Columbia, MD, USA) equipped with an MABPac HIC-butyl column (Fisher) was used to remove unconjugated mAb and also analyze the drug–antibody ratio (DAR) of ADC. The purified ADC was neutralized to pH 7.0 with 1 M Tris solution, sterilized, and mixed with 0.1% sodium azide for long-term storage at −80 °C. 

### 3.4. Flow Cytometry Analysis

The meningioma cell surface-binding of anti-SSTR2 mAb was analyzed using a BD LSRII flow cytometer (BD Biosciences, San Jose, CA, USA). The mAb was labeled with an Alexa Fluor™ 647 labeling kit to generate mAb-AF647. Approximately 1 × 10^6^ meningioma CH157-MN cells or normal AC07 cells were stained with 1 µg mAb-AF647 in 100 µL PBS at room temperature for 30 min in the dark, and washed with PBS before flow cytometry analysis [50]. 

### 3.5. In Vivo Imaging System (IVIS) Imaging

The growth of meningioma (CH157-MN-FLuc) tumors in xenografted NSG mice was monitored by measuring bioluminescent signal (FLuc) using an IVIS Lumina Series III (PerkinElmer, Waltham, MA, USA) every two days post cells injection. To monitor meningioma targeting, the anti-SSTR2 mAb or ADC was labeled with Cyanine 5.5 (Lumiprobe, Hunt Valley, MD, USA) according to the manufacturer protocol. The Cy5.5-labeled mAb or ADC was intravenously (i.v.) injected into mice via tail vein. At 24 h post injection, the xenograft mice were imaged under IVIS with a wavelength of 660/710 nm (excitation/emission) and an exposure time of 10 s to analyze the meningioma targeting and biodistribution in vivo. The important organs, including brain, heart, lung, kidney, and spleen, were also extracted to collect ex vivo images to check the possible off-target binding. 

### 3.6. In Vitro Anti-Meningioma Cytotoxicity

In the in vitro anti-cancer cytotoxicity assay [38], CH157-MN cells or normal arachnoidal AC07 cells were seeded in 96-well plates in triplication with a VCD of 50,000 cells/mL in 75 μL of DMEM/F12 complete growth medium. After 24 h incubation at 37 °C in a CO_2_ incubator, 75 μL of medium-containing drug was added into each well to reach final DM1 concentrations of 0.1–300 nM or ADC concentrations of 0.5–500 nM. After 72 h incubation, the anti-cancer cytotoxicity was measured using CellTiter-Glo Luminescent Cell Viability Assay (Promega, Madison, MI, USA). The relative viability was calculated using the detected luminescent signal which is proportional to the viable cell number. The IC_50_ value was calculated using the ED50V10 Excel add-in. 

### 3.7. Pharmacokinetics (PK)

The serum stability of ADC was investigated by i.v. administering 10, 15, 20, and 25 mg/kg into four groups of 6-week old NSG mice (*n* = 3). The serum samples were collected at 0.5, 2, 7, 24, 48, and 72 h and frozen at −80 °C for ELISA titration. The previously established PK modeling [39,49,51] was applied to calculate the area under the curve (AUC), half-life t_1/2_ = $\frac{0.693V_{d}}{CL}$, recommended dose (D) = $C_{max.desired}k_{e}V_{d}T\frac{1-e^{-k_{e}}}{1-e^{k_{e}}T}$, and recommended dosing interval (τ) = $\frac{\ln C_{max.desired}-lnC_{min.desired}}{k_{e}}+T.$ These PK parameters were used to decide the menginioma treatment strategy in the anti-tumor animal study.

### 3.8. In Vivo Anti-Meningioma Efficacy Study

When the detected bioluminescence intensity was over 1000 in IVIS imaging, the CH157-MN-FLuc intracranially xenografted NSG mice were randomized into 3 groups (*n* = 4), and i.v. administrated with 8 mg/kg anti-SSTR2 mAb, 8 mg/kg ADC, and 16 mg/kg ADC (empirically determined from the PK study) in 50 µL of saline via tail vein on Days 9 and 12. Two injections were conducted with an injection interval of 3 days during the treatment period until we observed slow locomotion and an obvious body weight drop in the control group on Day 15. Tumor volume was monitored with IVIS imaging every three days post meningioma implantation. 

### 3.9. Hematoxylin and Eosin (H&E) Staining

All tissue samples were embedded, sectioned at 5 μm, and mounted on frosted microscope slides (Fisher Scientific) for H&E staining. After dewaxing with xylene, the slides were hydrated with gradient ETOH, stained with hematoxylin and eosin Y solutions, dehydrated in absolute alcohol, cleared in xylene, and mounted with cytoseal Xyl. The stained slides were imaged with a high-performance Nikon microscope (Irving, TX, USA). 

### 3.10. Statistical Analysis

All statistical analysis was performed using GraphPad Prism. Two group comparisons were performed using unpaired Student’s *t* test to determine the probability of significance. Multiple comparisons were performed with ANOVA. The sample size in the animal study was determined following our previous ADC therapy study [39,50]. The *p* values were adjusted for multiple testing errors and ** *p* < 0.005 was considered as significant for all tests. All the experimental data were presented as mean ± standard error of the mean (SEM).

## 4. Conclusions

This study has developed and evaluated an anti-SSTR2 monoclonal antibody–drug conjugate for meningioma-targeted therapy. The constructed ADC significantly inhibited the meningioma tumor growth in an intracranial xenograft model. Importantly, the toxicity study and pharmacokinetics study did not detect adverse body weight, behavior changes, or histopathology. Taken together, the developed anti-SSTR2 ADC has a great potential to treat SSTR2-positive meningioma by the targeted delivery of a potent small molecule with minimal side effects. Despite the positive results, this study has some limitations to address in the future: (1) The full evaluation of toxicology and the treatment optimization of the SSTR2 ADC have not been performed; (2) The developed anti-SSTR2 mAb can be further engineered (such as by humanization) to meet the clinical requirement in the future; and (3) Alternative animal models such as the patient-derived xenograft (PDX) model or the humanized model should be developed to further evaluate the meningioma treatment efficiency. In the near future, we will further investigate factors relating to meningioma treatment with SSTR2 ADC, such as dosage optimization, survival, pharmacodynamics analysis, toxicology, and biodistribution, to collect translational data to facilitate the possible clinical evaluations. The correlation between the responses of benign meningioma and aggressive meningioma with the SSTR2 ADC will also be evaluated.

## Figures and Tables

**Figure 1 pharmaceuticals-14-00427-f001:**
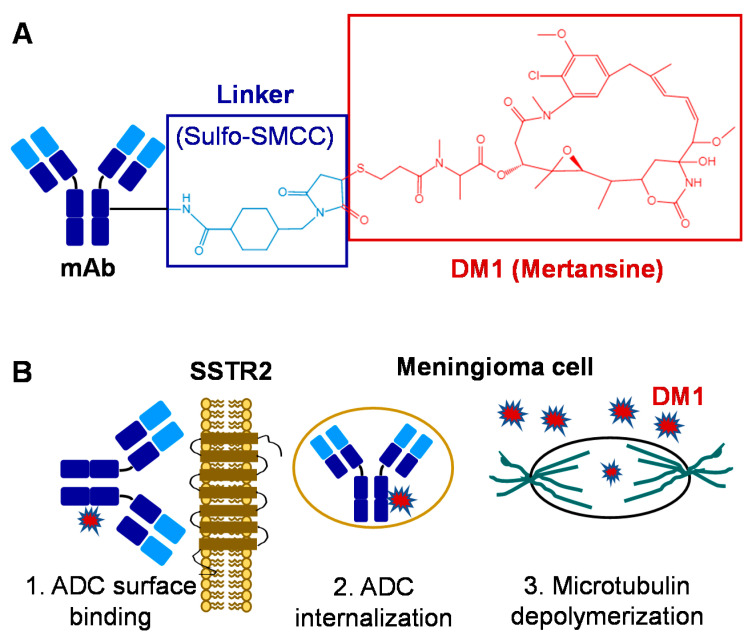
ADC-based targeted therapy for meningiomas. (**A**) Construction and structure of ADC: mAb is conjugated with mertansine (DM1) via sulfo-SMCC linker. (**B**) Mechanism of ADC to treat meningiomas: 1. Surface targets and binds the SSTR2 receptor overexpressed in meningiomas; 2. Internalizes and releases drug; 3. DM1 depolymerizes microtubulin and inhibits tumor cell proliferation.

**Figure 2 pharmaceuticals-14-00427-f002:**
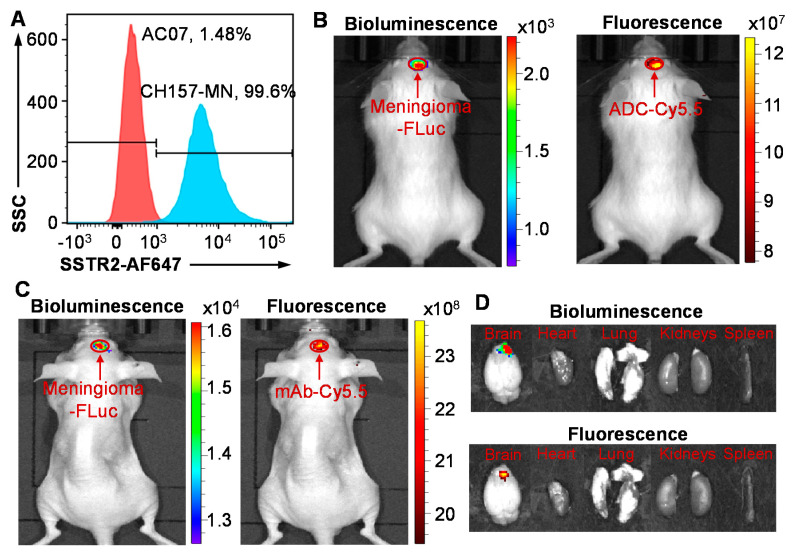
Meningioma targeting by anti-SSTR2 mAb and/or ADC. (**A**) Evaluation of the SSTR2 surface binding rate of mAb-AF647 in meningioma cell CH157-MN (blue) and normal arachnoidal cell AC07 (control, red) by flow cytometry analysis. Cells were stained with 1 μg of mAb-AF647 per million cells at room temperature for 30 min. (**B**) In vivo evaluation of specific targeting to meningioma (CH157-MN-FLuc) xenograft in NSG mice by anti-SSTR2 mAb-Cy5.5 using IVIS imaging. (**C**) In vivo meningioma targeting by anti-SSTR2 ADC-Cy5.5 in nude mice. (**D**) Ex vivo images of tumor and important organs including brain, heart, lung, kidney, and spleen. Total of 30 µg mAb-Cy5.5 or ADC-Cy5.5 was i.v. injected into mice (*n* = 3) through tail vein. Live-animal or ex vivo images were taken at 24 h post ADC injection.

**Figure 3 pharmaceuticals-14-00427-f003:**
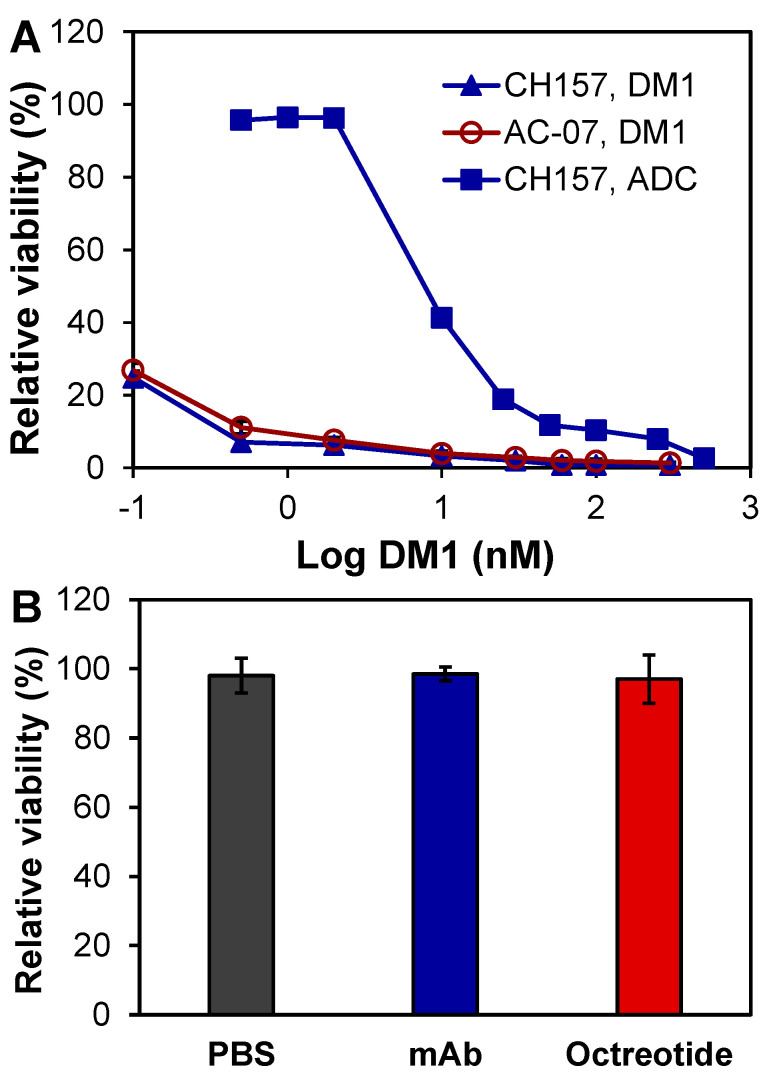
In vitro anti-meningioma cytotoxicity. (**A**) Evaluation of free drug and ADC using CH-157-MN and AC07 cells. (**B**) Effect of PBS, mAb, and octreotide (controls) in CH-157-MN cells. Data represent mean ± SEM, *n* = 3.

**Figure 4 pharmaceuticals-14-00427-f004:**
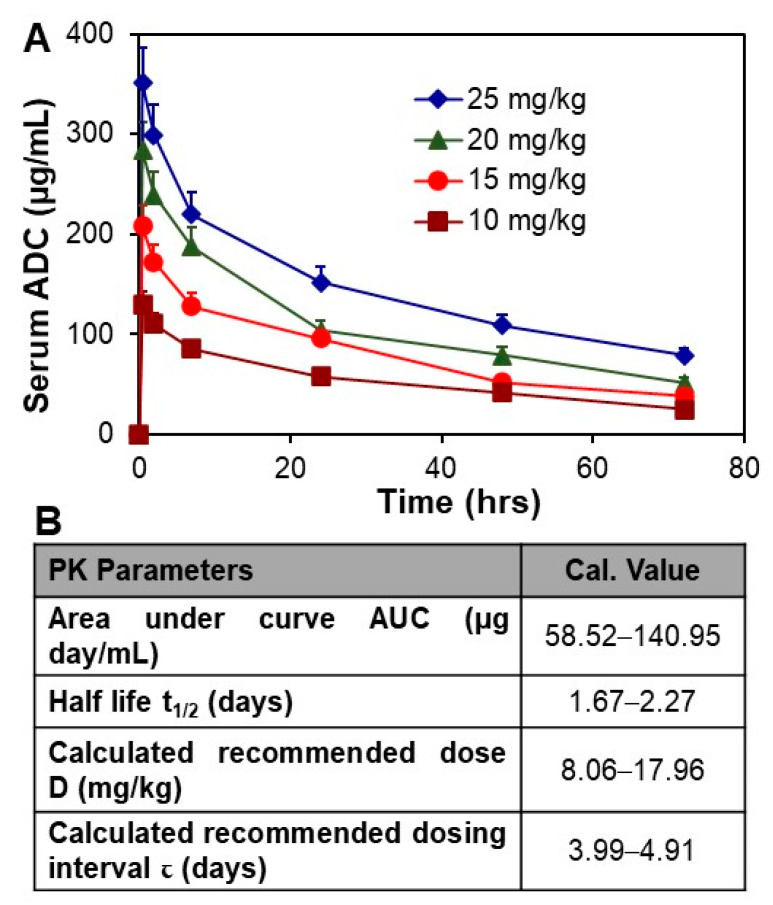
PK study of ADC. (**A**) Analysis of circulation stability by PK, *n* = 3. (**B**) The representative PK modeling parameters of ADC. Data represented as mean ± SEM.

**Figure 5 pharmaceuticals-14-00427-f005:**
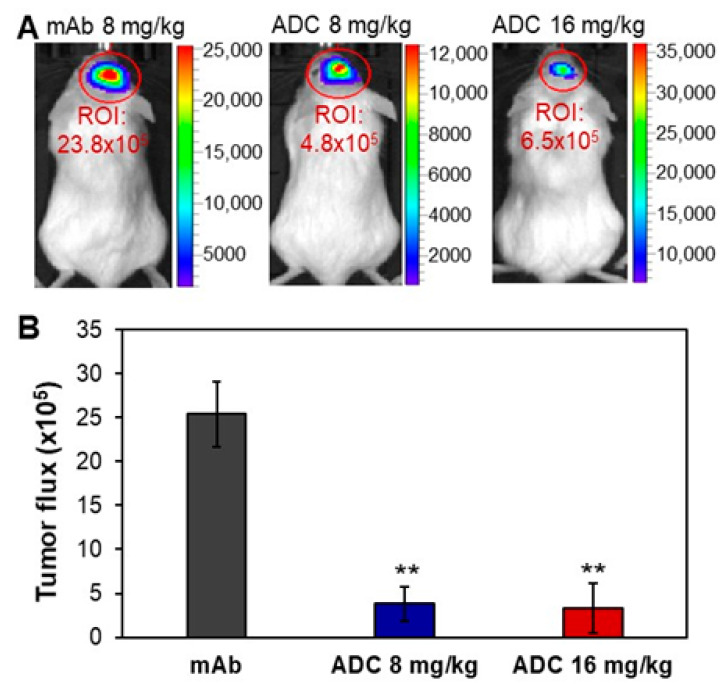
In vivo evaluation of the anti-meningioma efficacy of ADC. (**A**) Representative IVIS images of meningioma xenograft mice treated with mAb or ADC. (**B**) Tumor volume post treatment. The anti-SSTR2 mAb (8 mg/kg) or ADC (8 and 16 mg/kg) were administrated on Days 9 and 12 post intracranial xenograft. Tumor growth was monitored through measuring FLuc bioluminescence using IVIS. ** *p* < 0.005 vs. control using ANOVA followed by Dunnett’s *t*-test. Data represent mean ± SEM, *n* = 4.

**Figure 6 pharmaceuticals-14-00427-f006:**
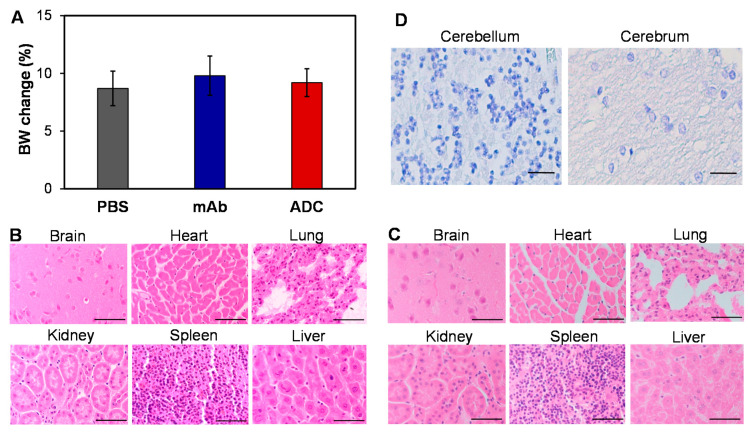
Evaluation of potential toxicity. (**A**) Body weight change of non-meningioma-carrying NSG mice post treatment with PBS, mAb, and ADC. (**B**,**C**) Evaluation of toxicity by H&E staining of important organs, including brain, heart, lung, kidney, spleen, and liver, in mAb- and ADC-treated groups. Scale bar equals 50 µm. (**D**) The high-resolution IHC images showed no binding of our anti-SSTR2 mAb to normal brain tissues (cerebellum and cerebrum). Scale bar equals 50 µm.

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
