# Peer review of "Antibody–Drug Conjugate to Treat Meningiomas"

_pharmaceuticals, 2021, doi:10.3390/ph14050427_

Round 1

Reviewer 1 Report

The authors present a discussion of their novel monoclonal antibody coupled Mertansine DM1 for treatment of aggressive, recurrent meningiomas. The antibody is directed against a somatostatin receptor highly expressed in meningiomas. Their approach makes use of the high expression of SSTR2 in meningiomas as a target for drug delivery, rather as a therapeutic target itself, as has been investigated previously with the use of somatostatin analogs. This endeavor is valuable as there are currently no chemotherapeutic agents in use that have been shown to significantly affect recurrence in this aggressive subtype of meningioma. The study is conducted using first in vitro analysis, followed by a mouse xenograft model. The methods are explained clearly and appropriate controls and toxicity evaluations have been completed. Overall this is a meaningful study that, though it’s applications are constrained by the perennial limitations of in vitro and xenograft models, suggests a potentially powerful new therapeutic modality for recurrent meningiomas that warrants further investigation.

            There are some minor points of clarification that need to be addressed prior to publication. These are the following:

“Introduction”

The introduction begins by stating the overall 5-year survival for meningiomas is 55-69% and citing “CBTRS 2012”. While the citation is not included in the reference section, the 2019 CBTRUS puts overall 5-year survival for malignant meningiomas at 66.5-69.9%, while the overall 5-year survival for non-malignant meningiomas 87.8-88.2%. The authors should specify whether they intend to discuss the survival for only malignant meningiomas, or meningiomas as a whole, as these statistics are very different. The appropriate source and citation should be used.

The introduction also states that, when it comes to somatostatin analogs, “treatment efficacy is debated in clinical practices because of its poor ability to control meningioma tumor growth”, but then cites three papers, including a review article, all of which offer an optimistic view on the efficacy of somatostatin analogs, though highlight the need for further study. This point needs to be further clarified to reflect the content of the cited literature.

“Results and discussion Section 2.6 Toxicity Evaluation”

The authors discuss the use of systemic therapies to control “metastatic meningiomas”. I believe they are referring to aggressively recurrent meningiomas. This should be phrased as such. Metastasis in meningioma is exceedingly rare and, while there is reference to a single case of metastatic disease in one of the cited sources, the papers cited are primarily discussing aggressive, recurrent disease, not metastasis.

There is no specific discussion of the limitations of the study, including those inherent to the xenograft model. This should be included.

Reviewer 2 Report

pharmaceuticals-1187545

In this study anti-SSTR2 antibody-drug conjugate (ADC) was tested and evaluated to treat meningiomas. The evaluation was performed on cell lines and on an intracranial xenograft mouse model. The comparisons were made using the normal arachnoidal AC07 cells. After surface binding, the ADC is internalized into the cytoplasm of meningioma cells through receptor-mediated endocytosis to form a late endosome, and free drug is released via lysosomal degradation. Extensive testing was performed and the authors concluded that the anti-SSTR2 ADC can target and stop meningioma tumor growth. The study showed that the anti-SSTR2 mAb had a high binding rate to meningioma cells but low binding to the normal arachnoidal cells. The In Vivo Imaging System (IVIS) demonstrated that the labelled ADC accumulated in meningioma xenograft but not in normal organs. The pharmacokinetics study and histological analysis confirmed its stability and minimal toxicity.

The English in the paper could be improved and several parts need to be rewritten. The paper has specific parts that need to be revised.

First of all, in the abstract the sentence: “As the most common primary tumors of the central nervous system, proliferative meningiomas are ag-gressive and recurrent“, should be reorganized because its meaning is confusing. Proliferative aggressive meningiomas are not the most common primary tumors of the CNS, meningioma on the whole are, but grades II and III are much less frequent. Please rephrase (first sentences of the Introduction on page2 are fine explaining frequencies).

In the abstract there are many typos especially hyphens in the words for example ag-gressive or imag-ing etc.

Also the last sentence of the Abstract This study demonstrated that the anti-SSTR2 ADC can target and control meningioma tumor growth. The word control is confusing control how? It would be better to state if it reduces the growth or stops the growth.

Introduction page 2, please explain what (CBTRS 2012) means or include citation or web link.

Also first sentence meningiomas are ….tumors plural.

Introduction page 2, „The tumors contain widespread genomic disruption (amplification,

deletion and rearrangement), NF2 and other mutations [3,4].“ This sentence should be reorganized since it is not understandable and not to confuse gene and syndrome to: ….of the NF2 gene and mutations in other relevant genes.

The following paragraph should be rewritten to be more clear and easier to read: „Several clinical studies showed that it is safe to target SSTR2 [6,9,10], but numerous clinical trials have failed to identify systemic medical therapies, such as sunitinib and everolimus/octreotide, to effectively control tumor growth [11-13]. For example, a clinical trial that tests the combination of everolimus with dosage of 10 mg/day for 28 days and octreotide with dosage of 30 mg/day for one day shows 50% tumor growth inhibition in 16 patients [13]. Although somatostatin analog has been used in diagnosis and detection [14,15], its treatment efficacy is debated in clinical practice because of the poor ability to control meningioma tumor growth [7,16,17]. It is imperative to develop a targeted therapy to treat low and high-grade meningiomas.“ Furthermore, insert words ....to treat the growth of both low- and high-grade meningiomas.

My main concern of the paper is that it is not clear if the authors have constructed the ADC in this contribution or here they just develop and evaluate it and it was constructed before. This is unclear from the Introduction where reference 35 was mentioned by the same authors Ou, J.; Si, Y.; Goh, K.; Yasui, N.; Guo, Y.; Song, J.; Wang, L.; Jaskula-Sztul, R.; Fan, J.; Zhou, L., et al.  Bioprocess development of antibody-drug conjugate production for cancer treatment. PLoS One 2018, 13, e0206246, doi:10.1371/journal.pone.0206246. Reference 36 too.

The same applies for the Results section where again it is not clear from the following paragraph: „In this study, we constructed and evaluated the SSTR2-targeting ADC for meningioma treatment. We have previously developed an anti-human and mouse SSTR2 mAb (IgG1 kappa) that targets the extracellular domain of SSTR2 using hybridoma technology [36].“ Please indicate clearly if this is constructed now or previously.

On page 3, Figure 1 please indicate DM1 drug which is released also indicate drug in the Figure itself and the full name in the legend.

Another point - it would be interesting to investigate some kind of correlation between benign meningioma and aggressive malignant cases in the application of the anti-SSTR2 antibody-drug conjugate (ADC).

There could be more references. Please include additional reference in the Introduction: Jovčevska I, Muyldermans S. The Therapeutic Potential of Nanobodies. BioDrugs. 2020 Feb;34(1):11-26. doi: 10.1007/s40259-019-00392-z. PMID: 31686399; PMCID: PMC6985073.

The paper is furnished with many figures. In my opinion this is a valuable contribution that deserves to be published. My recommendation to the authors is to incorporate major revisions.

Reviewer 3 Report

The authors have engineered an antibody drug conjugate (ADC) to treat meningiomas. This ADC comprises of a somatostatin receptor (SSRT) 2 mAb conjugated to Mertansine. As known from previous studies SSRT2 is highly expressed in meningiomas and is a clinically relevant target. The anti-SSR2 ADC binds to SSR2 expressing meningiomas and undergoes receptor-mediated endocytosis and then free DM1 is released via lysosomal degradation. The released DM1 inhibits microtubules and causes apoptosis in meningiomas. The authors have evaluated the anti-SSRT2 ADC’s binding affinity to meningioma cell lines in vitro, its pharmacokinetics and potency of orthotopic meningioma inhibition in vivo and performed toxicity profiling. Treatment of meningiomas is an unmet need and more novel treatment modalities are required. The experimental approach of the authors is good, however there are some concerns that the authors need to address.

1) The abstract states that the authors have used 4mg/kg and 8 mg/kg body weight of anti-SSRT2 ADC to inhibit tumor growth. However the results show the usage of 8 mg/kg and 16 mg/kg. Which one is correct?

2) In Figure 5B, the tumor flux does not change between 8mg/kg and 16mg/kg , however in Fig 5A, the yellow and red intensity of the flux is decreased in 16 mg/kg as compared to 8 mg/kg. Why does the quantification not reflect what we observe in the figure?

3) Also could the authors speculate why 16 mg/kg dose does not inhibit tumor growth more than 8mg/kg, since a dose dependent effect of the ADC is observed in vitro (Fig 3A)

4) Why do the authors not see an improvement in overall survival of the mice when treated with the ADC?

5) Could the authors state the PK parameters as calculated in Fig 4B and plasma stability, for octreotide combined with Lutathera explicitly so the readers would know how the anti-SSR2 ADC engineered by the authors compared to the former and how would it be beneficial.

6) The authors should either show an apoptotic marker staining of tumors in vivo OR cite the literature that has shown that anti-SSR2 ADCs inhibit tumor growth via apoptosis.

Round 2

Reviewer 3 Report

The authors have addressed my concerns adequately. I therefore recommend the manuscript for publication.